# Ten Years of ECHO Chronic Pain and Opioid Stewardship in Ontario: Impact and Future Directions

**DOI:** 10.3390/healthcare13243203

**Published:** 2025-12-08

**Authors:** Andrea D. Furlan, Q. Jane Zhao, Paul Taenzer, Andrew J. Smith, Ralph Fabico, Kiera Morgan, Rhonda Mostyn, John F. Flannery

**Affiliations:** 1KITE Research Institute, University Health Network, Toronto, ON M5G2A2, Canada; 2Department of Medicine, University of Toronto, Toronto, ON M5S 3H2, Canada; 3Institute for Work & Health, Toronto, ON M5G 2E9, Canada; 4Department of Physical Medicine and Rehabilitation, Queens University, Kingston, ON K7L 3N6, Canada; 5Pain and Addiction Medicine, Centre for Addiction and Mental Health (CAMH), Toronto, ON M6J 1H4, Canada

**Keywords:** chronic pain, health professions education, continuing professional development, knowledge mobilization, program evaluation

## Abstract

**Highlights:**

**What are the main findings?**
Over 10 years, our hub has trained more than 1500 healthcare professionals from across Ontario.Evaluations of the impact of Extensions for Community Healthcare Outcomes (ECHO) Pain showed positive results among healthcare professionals using both quantitative and qualitative research methods.

**What are the implications of the main findings?**
Implementation of a new program takes a team effort.Sustainability of a new program involves infrastructure, adequate staff, IT support, access to experts, recruitment of participants, and case presentations.

**Abstract:**

**Background**: ECHO Pain is a health professions education model that uses telehealth technology to connect specialists in academic centres to healthcare professionals who work in the community to disseminate best practice knowledge and foster interprofessional collaboration to support real patient cases. **Methods**: This paper summarizes 10 years of ECHO Pain implementation and evaluation in Ontario. We reviewed participants’ demographics, characteristics of cases presented in ECHO sessions, and the research output of this ECHO Pain program. **Results**: From June 2014 to June 2024, there were 529 sessions, 1527 healthcare professionals from urban and rural regions attended ECHO, and 25,898 h of continuing medical education credits were provided. We published 11 papers in peer-reviewed scientific journals using qualitative and quantitative research methods. **Conclusions**: ECHO Pain has been implemented and sustained in Ontario for 10 years, with demonstrated interprofessional education and an ongoing community of practice to discuss chronic pain cases. ECHO Pain is filling a significant gap in health professions education related to chronic pain in Ontario, especially for primary care professionals living in rural, remote, and underserved areas.

## 1. Introduction

Chronic pain affects one in five Canadians and is a leading cause of disability and poor quality of life [1]. In Ontario, people with chronic pain utilize CAD 1742 more per year in healthcare than people who do not have chronic pain [2], and more than half of all opioid related deaths still involve prescription drugs (either dispensed or diverted) [3]. Most patients with chronic pain in Ontario are managed by primary care health professionals, with little or no access to specialists, long wait times to multidisciplinary programs or pain clinics, insufficient opioid prescribing skills, limited interprofessional education and practice, and difficulties in managing the burden of mental health and complex social needs [4].

ECHO (Extensions for Community Healthcare Outcomes) is a health professions education model that uses telehealth technology to connect specialists in academic centres to healthcare professionals who work in the community, with the goal of disseminating best practice knowledge and fostering interprofessional collaboration to support real patient cases. In 2003, ECHO was developed for managing hepatitis C in New Mexico and has expanded to other medical conditions such as chronic pain, mental health disorders, substance use disorders, diabetes, and rheumatological conditions, among others. There is accumulating scientific evidence of it being an effective method to transfer knowledge and skills and create a learning community where all learn and all teach [5,6,7]. In 2014, our team implemented the first ECHO program in Canada, for chronic pain and opioid stewardship, as a collaboration between the University Health Network (UHN) and Queen’s University.

The main goals of Project ECHO Chronic Pain and Opioid Stewardship (ECHO Pain) are to disseminate knowledge, improve skills, and foster a community of practice among specialists and primary healthcare professionals to improve care for people with chronic pain. The four core tenets of the ECHO model are (1) to use technology to leverage scarce healthcare resources, (2) share best practices and reduce variation in care; (3) harness practice-based learning and develop expertise among healthcare professionals, and (4) monitor and evaluate outcomes of the ECHO model, and, when indicated, adopt changes to improve the desired outcomes [8].

The objective of this study is to describe the implementation activities and results of ten years of the UHN/Queens ECHO Chronic Pain and Opioid Stewardship (ECHO Pain) in Ontario, Canada.

## 2. Materials and Methods

Our ECHO Pain program started offering weekly sessions in June 2014. An ECHO session is structured in two parts: case presentation and a short didactic lesson. The healthcare professionals who register to attend weekly sessions are invited to present a de-identified patient case. The session is moderated by a member of the expert hub. The professions represented in the expert hub include medicine (family medicine, physiatry, addiction medicine, psychiatry, and neurology), pharmacy, psychology, physiotherapy, occupational therapy, chiropractic, nursing, social work, and health sciences librarianship. Since 2023, the expert hub has been complemented by a person with lived experience of chronic pain.

The case discussion follows a sequence of (1) 5 min case presentation, (2) questions of clarification by the participants, (3) questions of clarification by the expert hub members, (4) recommendations by the participants, and (5) recommendations by the expert hub members. The second part of the session is a short 20 min didactic by one of the hub members. The curriculum of didactic presentations is repeated with minor modifications at every cycle of 20 to 21 sessions, and it is accredited for both specialists and family physicians. Our ECHO hub offers 40 to 42 sessions per year, divided into cycles of approximately 20 to 21 sessions.

The ECHO hub is supported by a strong infrastructure that includes a project manager, educational coordinator, research coordinator, program coordinator, administrative assistant, and IT support. In the initial phases of this ECHO Pain program, the technology used for videoconferencing was the Ontario Telemedicine Network (OTN), but in 2018, it was replaced by Zoom^®^ (version Zoom 4.5).

As one pillar of the ECHO model is outcome measures and quality improvement, we have a dedicated research coordinator to collect data, monitor outcomes, and conduct research. With funding from various grants, we were able to recruit summer and graduate students to conduct research on the ECHO Pain activities.

We conducted a review of the ECHO Pain implementation, participants’ registrations, and the cases discussed from June 2014 to June 2024. We summarized the data using descriptive statistics.

Funding for this program came from a diverse range of sources. Initially, we obtained a Canadian Institutes of Health Research (CIHR) planning grant that allowed our group to visit the University of New Mexico in 2013 to obtain training to develop and implement our own ECHO program. Crucial to the start of ECHO in Canada was the pilot funding from the Ontario Ministry of Health and Long-Term Care (MOHLTC), an Ontario Medical Association (OMA) grant, and the Northern Ontario Academic Medicine Association (NOAMA) grant to expand ECHO pain to Northern Ontario.

The research activities were funded by a CIHR Partnerships in Health System Improvement (PHSI) grant and a CIHR Opioid Crisis grant to examine two ECHO programs that focus on chronic pain and opioids (Toronto and Thunder Bay). In 2017, we received funding from Health Canada, through the Substance Use and Addictions Program (SUAP), to spread ECHO Pain to other Canadian Provinces. In 2023, we were funded by CIHR in the Transforming Health with Integrated Care (THINC) Implementation Science Team Grant initiative, to use implementation science methods to evaluate the spread and scale of ECHO Pain in Canada.

## 3. Results

### 3.1. Program Activities and Outputs

From June 2014 to June 2024, over ten years of offering ECHO, we conducted a total of 529 ECHO Pain sessions, which were delivered in cycles of 20 to 21 sessions (January to June and August to December). The sessions’ duration was initially 2 h and was reduced to 90 min beginning in 2020 (cycle 12). The sessions are currently offered on Thursdays from 12:30 to 2:00 p.m.

### 3.2. Reach and Participation

#### 3.2.1. Patients Reached

There were 529 cases presented, of which 483 were new and 46 were follow-up cases. The mean age of patients presented was 54.0 years (standard deviation of 15.8 years), with a range from 12 to 101 years old. The cases discussed involved 302 (57%) females, 199 (38%) males, and 28 (5%) unknown gender cases. Information about country of birth was available in 462 cases, of which 363 (79%) were born in Canada, and the remaining were distributed widely from around the world.

The types of pain diagnoses were documented in 500 cases, and the most common were back pain in 290 (58%), fibromyalgia in 115 (23%), arthritis in 110 (22%), and headaches in 94 (19%). Although not so common, we discussed 18 cases (4%) of complex regional pain syndrome (CRPS).

Comorbidities were documented in 489 cases, of which the most common were depression in 216 (44%), sleep disorder in 175 (36%), substance use disorder in 138 (28%), anxiety in 134 (27%), history of trauma in 89 (18%), and other mental health condition in 136 (28%).

There were 356 (67%) cases involving discussions around opioids for chronic pain, such as appropriate indications for treatment with opioids, adequate dosage, risk assessment, diagnosis of opioid use disorder, polypharmacy, tapering, and switching opioids.

#### 3.2.2. Healthcare Professionals’ Participation

There were 1527 healthcare professionals who attended ECHO Pain sessions during this period, of which 1422 (93%) were from Ontario, 64 (4%) were from other Canadian provinces, and 41 (3%) were international. The average number of participants per session was 34.8 people. The number of continuing medical education (CME) credits provided during this period was 25,898 h.

Participants’ professions were obtained from 1423 participants, and they are summarized in Table 1. The top five professions included family physicians (28.7%), nurse practitioners (25.1%), pharmacists (13%), registered nurses (10.5%) and social workers (3.9%).

Participants came from all Ontario Health regions of Ontario: (1) North East, (2) North West, (3) East, (4) Central, (5) Toronto, and (6) West (Figure 1). Most notably, participants came from southern Ontario, where the majority of the Ontario population resides.

### 3.3. Indicators of Effective Implementation

Over the past ten years, we have published 11 peer-reviewed articles related to the work of ECHO Pain at UHN/Queen’s. Below, we summarize the scientific articles that demonstrate effective implementation.

#### 3.3.1. Qualitative Findings

A qualitative study of three focus groups of ECHO participants in Ontario showed that managing patients with chronic pain in primary care can be difficult, particularly in remote or underserved practices. ECHO Pain was found to offer guidance to primary care professionals for their most challenging patients, promote knowledge acquisition and diffusion, and foster a “community of practice” [9]. Another qualitative study examined how ECHO promotes an interprofessional approach to chronic pain management. This study used focus group discussions with professionals from various healthcare disciplines, who shared mainly positive views of ECHO’s emphasis on interprofessional care [10]. A qualitative study using in-depth semi-structured phone interviews was conducted to obtain primary care professionals’ experiences and perceptions of participating in ECHO. The study explored the motivations for attending a chronic pain tele-mentoring program, as well as the effects of their participation on knowledge and interprofessional approach to chronic pain [11]. Lastly, we conducted a study to investigate patients’ perspectives on the pathway from primary care professionals’ presentations of their cases to their chronic pain management. Using data from in-depth interviews with 20 patients, along with their associated case presentation forms and the recommendations following the presentations, we examined the alignment of patient and practitioner views and perceptions of how ECHO affected them. We found that the impact on patients is indirect but positive: most respondents expressed pleasure in contributing to research about chronic pain management, though only two of them identified a direct impact on their own treatment. They also appreciated their practitioner’s efforts to bring expert attention to their patient’s situation [12].

#### 3.3.2. Quantitative Findings

We analyzed the results of pre- and post-ECHO questionnaires from 170 participants in the first three years of ECHO Pain operation and showed that ECHO improved participants’ self-efficacy and knowledge. Self-efficacy improvement was significantly higher among physicians, physician assistants and nurse practitioners than the non-prescribers group. Moreover, satisfaction with the program was highly rated [13]. We also conducted an administrative database analysis in Ontario comparing opioid prescribing patterns between family doctors who attended with family doctors who did not attend ECHO Pain. We found that physicians who were already high-dose opioid prescribers self-selected to participate in the ECHO program, demonstrating that ECHO is reaching the clinicians who need it most. This study also showed that there was a significant reduction in high-dose opioid prescription in the ECHO group compared to a non-significant increase in the matched controls [14].

#### 3.3.3. Sustainable Program Funding

The ECHO Pain program has continued receiving sustainable funding from the Ontario Ministry of Health from 2014 to the present date.

## 4. Discussion

These results illustrate the implementation activities and outputs of the first ECHO hub in Canada, with sustained activities over a 10-year period. Our hub has trained more than 1500 healthcare professionals from across Ontario. The types of chronic pain patients discussed during ECHO weekly sessions reflect the most common pain conditions seen in primary care, such as patients with back pain, fibromyalgia, headaches, mental health comorbidities, and/or opioid-related cases. Our expert hub has maintained a wide variety of health professions across eight disciplines over the 10-year period. Recruitment of participants and cases has been challenging, but we were able to sustain attendance and cases for discussion.

Evaluations of the impact of ECHO Pain showed positive results among healthcare professionals, using both qualitative and quantitative research methods. The top takeaways from our research are as follows: (1) ECHO Pain changes clinical behaviour—both quantitative and qualitative data show that clinicians’ confidence and knowledge related to pain and opioid management increase and that ECHO Pain fosters a strong community of practice [9,10,11,12,13]; (2) ECHO Pain attracts those who need it the most—high opioid prescribing physicians not only self-select to attend but also prescribe fewer opioids than their peers after attending ECHO [14]; and (3) ECHO Pain is equitable—through use of telehealth technology, ECHO Pain provides timely education to clinicians practising in rural, remote, and underserved communities [11,14].

Interprofessional education initiatives remain challenging to implement and require a well-designed pedagogical process [15]. Our qualitative research showed that ECHO Pain fills an important gap by creating and fostering a community of practice where healthcare professionals from various disciplines and professions can share their knowledge, skills, and resources, therefore achieving the goal of ECHO, which is to democratize knowledge. ECHO has been used around the world to reduce barriers to the training and education of healthcare professionals in a wide variety of topics.

Collection of patient outcomes for any health professions education program has historically been difficult. While improving patient care and outcomes is an important downstream goal, education programs need to have enough capacity in terms of funding and human resources to be able to collect these outcomes. One recent scoping review of the ECHO literature examined publications from January 2003 to June 2020 for the reporting of patient or community health outcomes. The authors only identified 15 publications that report patient outcomes and 1 publication mentioning community health outcomes [16].

From our experience collecting patient health outcomes, this research was difficult to conduct for several reasons: (1) It was difficult to consent patients to the research study as we only have direct access to their primary care professionals; (2) depending on the patient population, collecting consent, conducting research, and maintaining low drop-out rates may be difficult due to the patient’s unstable health and social conditions; and (3) changing health outcomes in some patient populations is difficult as patients are extremely complex and health outcome change may occur on a longer timeframe than the study duration. Factors that contributed to the success of ECHO Pain collecting patient outcomes include having a dedicated staff member, the research coordinator, to collect this data, having shorter-term students whose projects were related to this research, and ongoing funding for the research study, which helped pay study participants, research students, transcription costs, and other research costs. Ultimately, it was worthwhile to hear patients’ own voices regarding their experiences of chronic pain and the impact of ECHO Pain for them, but the path to achieving these outcomes was time and cost-intensive [12].

Sustainability is an important topic for ECHO Pain, and it involves infrastructure, adequate staff, IT support, access to experts, recruitment of participants, and case presentations. Our ECHO Pain program evolved over the 10-year period by applying quality improvement methods, feedback from participants, regular consultation with an advisory board committee, peer-reviewed grants and publications, presentations at conferences, and reporting to the funders.

### Future Directions: Implementation and Evaluation Lessons for Future ECHO Programs

Over the past decade, we have learned that implementing and sustaining an ECHO program depends on more than just subject matter expertise and telehealth technology. Implementing a successful and sustained ECHO program requires robust and coordinated infrastructure, including dedicated project management, research and educational support, IT resources, and engagement with a multidisciplinary hub team. One of the most important lessons for new implementers is the need for consistent recruitment of both participants and patient cases, which can be a persistent challenge. We also found that maintaining engagement requires careful attention to the relational aspects of ECHO—cultivating a community of practice where knowledge flows in multiple directions.

From an evaluation perspective, integrating research into the program’s design from the beginning was essential for demonstrating value to funders and stakeholders. Our multimethod approach allowed us to understand not just whether ECHO worked, but how and for whom it was most effective. We focused on clinician self-efficacy, prescribing practices, interprofessional collaboration, and equity of access as key outcomes of interest. However, measuring patient-level outcomes remains resource-intensive and methodologically complex. Future programs should anticipate these challenges and build evaluation capacity early, including allocating dedicated research personnel and pursuing funding opportunities aligned with both service delivery and scholarly inquiry.

The hub didactics and quality improvement activities are expected to evolve as current themes in healthcare are reflected in the curriculum. In Canada, a network of ECHO Pain is being developed and fostered, with more ECHO Pain hubs gaining research support from CIHR THINC grants. As a result, a growing number of different ECHO Pain hubs can share resources and lessons learned.

## 5. Conclusions

ECHO Pain has been implemented and sustained in Ontario for 10 years, with demonstrated interprofessional education and an ongoing community of practice to discuss complex chronic pain cases. By fostering case-based learning, ECHO promotes shared experiences and the development of core interprofessional competencies, such as effective communication and understanding of diverse professional roles. ECHO Pain is filling a significant gap in health professions education related to chronic pain in Ontario and better equipping the workforce to meet the needs of Ontario’s patient population, especially for primary care professionals living in rural, remote, and underserved areas.

## Figures and Tables

**Figure 1 healthcare-13-03203-f001:**
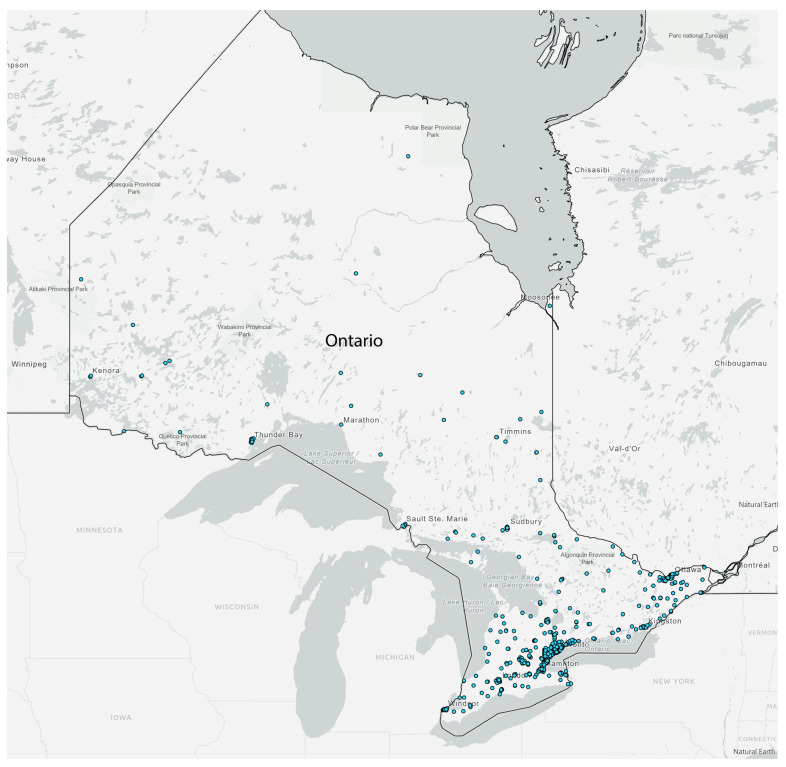
Geographical location of ECHO participants in Ontario.

**Table 1 healthcare-13-03203-t001:** Participants’ professions.

Professions	*n*	%
MD Family Physician	409	28.7%
Nurse Practitioner	357	25.1%
Pharmacist	185	13.0%
Registered Nurse	150	10.5%
Social Worker	55	3.9%
Occupational Therapist	51	3.6%
Chiropractor	37	2.6%
Physiotherapist	37	2.6%
MD Specialist	32	2.2%
Kinesiologist	20	1.4%
Physician Assistant	17	1.2%
Administrator	17	1.2%
Psychologist	17	1.2%
Registered Psychotherapist	3	0.2%
Registered Massage Therapist	3	0.2%
Chiropodist	2	0.1%
Other (count of 1 each)	31	2.2%

## Data Availability

Data available are upon request due to restrictions of the ethics committee.

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
