# Peer review of "Ten Years of ECHO Chronic Pain and Opioid Stewardship in Ontario: Impact and Future Directions"

_healthcare, 2025, doi:10.3390/healthcare13243203_

Round 1
Reviewer 1 Report
Comments and Suggestions for Authors
Dear Authors: I read your manuscript with interest. I applaud the ECHO Project work and was there when Dr. Ruth Dubbin first presented the concept many years ago. I enjoyed reading the culmination and development of this work into its present state by Dr. Furlan and co-authors more than a decade later. The manuscript is very well written and deserves to be disserminated in the scientific literature. I only have a few edits:
Line 153: change 'it is' to 'they are'.
Line 237: delete extra space inbetween 'the' and 'goal'.
References: Capitalize initials for Ontario, Canada, Canadian, Project ECHO, and initials of Journal names. Pain Medicine, etc.
Author Response
|
Dear Authors: I read your manuscript with interest. I applaud the ECHO Project work and was there when Dr. Ruth Dubbin first presented the concept many years ago. I enjoyed reading the culmination and development of this work into its present state by Dr. Furlan and co-authors more than a decade later. The manuscript is very well written and deserves to be disserminated in the scientific literature. I only have a few edits: |
Thank you |
|
Line 153: change 'it is' to 'they are'. |
Done |
|
Line 237: delete extra space inbetween 'the' and 'goal'. |
Done |
|
References: Capitalize initials for Ontario, Canada, Canadian, Project ECHO, and initials of Journal names. Pain Medicine, etc. |
The software that we use to format the references (EndNote) decapitalize all words in the titles. We downloaded the MDPI style recommended by this journal for references in Endnote. |
Reviewer 2 Report
Comments and Suggestions for Authors
Thank you for the opportunity to review this oversight article about the ten-year use of the ECHO approach. The aim was to describe the implementation activities and results of ten years of the UHN/Queens ECHO Chronic Pain and Opioid Stewardship (ECHO Pain) in Ontario, Canada.
The title, abstract, and introduction are well written and effectively explain the subject.
The methods section is also acceptable, explaining the different approaches used.
Results. I think the results paragraph should focus more on the implementation part. All the results from the different studies can be read elsewhere, and I believe that to justify this publication, there must be a greater focus on something new that has not been published elsewhere. Paragraph 3.3 is a self-referencing explanation of the results, offering nothing new. Maybe you can put these results in a table and then focus on the implementation. You state in the conclusion of the abstract that the ECHO method is successfully implemented. It could be interesting to know what kind of framework and approaches are used, what you considered to be a success criterion, and how you measured that. Please use the majority of the result section to elaborate on that subject.
Discussion. I don´t think you can conclude that it is well implemented. Please focus on lines 267-290, this is, in my opinion, the interesting part.
Author Response
|
Thank you for the opportunity to review this oversight article about the ten-year use of the ECHO approach. The aim was to describe the implementation activities and results of ten years of the UHN/Queens ECHO Chronic Pain and Opioid Stewardship (ECHO Pain) in Ontario, Canada. The title, abstract, and introduction are well written and effectively explain the subject. The methods section is also acceptable, explaining the different approaches used. |
Thank you. |
|
Results. I think the results paragraph should focus more on the implementation part. All the results from the different studies can be read elsewhere, and I believe that to justify this publication, there must be a greater focus on something new that has not been published elsewhere. Paragraph 3.3 is a self-referencing explanation of the results, offering nothing new. Maybe you can put these results in a table and then focus on the implementation. You state in the conclusion of the abstract that the ECHO method is successfully implemented. It could be interesting to know what kind of framework and approaches are used, what you considered to be a success criterion, and how you measured that. Please use the majority of the result section to elaborate on that subject. |
Thank you for these comments.
We changed the Results section to show the implementation outcomes as follows: - Program activities and outputs to demonstrate that ECHO Pain was actually done and how active it was -Reach and participation to demonstrate that the ECHO program was adopted - Indicators of effective implementation that explicitly connect data to effectiveness
We removed some references to studies that we published and were mostly descriptive, narrative papers. We left only the papers that showed implementation outcomes. |
|
Discussion. I don´t think you can conclude that it is well implemented. Please focus on lines 267-290, this is, in my opinion, the interesting part.
|
We removed the word “successfully” implemented from the abstract, results and discussion. We added a section in the results that explains we have obtained sustainable funding from 2014 to the present date. The ECHO Pain program is part of the global hospital budget that UHN receives every year. |
Round 2
Reviewer 2 Report
Comments and Suggestions for Authors
I think you have corrected the manuscript in an acceptable way